GSDMD suppresses keratinocyte differentiation by inhibiting FLG expression and attenuating KCTD6-mediated HDAC1 degradation in atopic dermatitis

Zhong Yi 1
Huang Taoyuan 2
Li Xiaoli 1
Luo Peiyi 1 luopeiyi@gwcmc.org
Zhang Bingjun 3 zhangbj5@mail.sysu.edu.cn
1 Department of Dermatology, Guangzhou Women and Children’s Medical Center , Guangzhou , China
2 Department of Dermatology, Dermatology Hospital of Southern Medical University , Guangzhou , China
3 Department of Neurology, The Third Affiliated Hospital of Sun Yat-sen University , Guangzhou , China
Uversky Vladimir
Electronic publication date: 2024 Jan 16
Publication date: 2024
Volume: 12
Electronic Location ID: e16768
Received 2023 Jul 14; Accepted 2023 Dec 15
Copyright: © 2024 Zhong et al.
Copyright year: 2024
Copyright holder: Zhong et al.
License: This is an open access article distributed under the terms of the Creative Commons Attribution License, which permits unrestricted use, distribution, reproduction and adaptation in any medium and for any purpose provided that it is properly attributed. For attribution, the original author(s), title, publication source (PeerJ) and either DOI or URL of the article must be cited.
License URL: https://creativecommons.org/licenses/by/4.0/

Keywords: Keratinocyte differentiation, Atopic dermatitis, Filaggrin, HDAC1, KCTD6, GSDMD

Funding: The Doctoral Scientific Research Startup Fund of Guangzhou Women and Children’s Medical Center TE-104(1) Basic Research Joint Project of Guangzhou Technology Bureau and Hospital 202102010161 This study is supported by the Doctoral Scientific Research Startup Fund of Guangzhou Women and Children’s Medical Center (No. TE-104(1)) and the Basic Research Joint Project of Guangzhou Technology Bureau and Hospital (No. 202102010161). The funders had no role in study design, data collection and analysis, decision to publish, or preparation of the manuscript.

==============================
Background

Recent studies have shown that activated pyroptosis in atopic dermatitis (AD) switches inflammatory processes and causes abnormal cornification and epidermal barrier dysfunction. Little research has focused on the interaction mechanism between pyroptosis-related genes and human keratinocyte differentiation.

Methods

The AD dataset from the Gene Expression Omnibus (GEO) was used to identify differently expressed pyroptosis-related genes (DEPRGs). Hub genes were identified and an enrichment analysis was performed to select epithelial development-related genes. Lesions of AD patients were detected via immunohistochemistry (IHC) to verify the hub gene. Human keratinocytes cell lines, gasdermin D (GSDMD) overexpression, Caspase1 siRNA, Histone Deacetylase1 (HDAC1) siRNA, and HDAC1 overexpression vectors were used for gain-and-loss-of-function experiments. Regulation of cornification protein was determined by qPCR, western blot (WB), immunofluorescence (IF), dual-luciferase reporter assay, co-immunoprecipitation (Co-IP), and chromatin immunoprecipitation (ChIP).

Results

A total of 27 DEPRGs were identified between either atopic dermatitis non-lesional skin (ANL) and healthy control (HC) or atopic dermatitis lesional skin (AL) and HC. The enrichment analysis showed that these DEPRGs were primarily enriched in the inflammatory response and keratinocytes differentiation. Of the 10 hub genes identified via the protein-protein interaction network, only GSDMD was statistically and negatively associated with the expression of epithelial tight junction core genes. Furthermore, GSDMD was upregulated in AD lesions and inhibited human keratinocyte differentiation by reducing filaggrin (FLG) expression. Mechanistically, GSDMD activated by Caspase1 reduced FLG expression via HDAC1. HDAC1 decreased FLG expression by reducing histone acetylation at the FLG promoter. In addition, GSDMD blocked the interaction of Potassium Channel Tetramerization Domain Containing 6 (KCTD6) and HDAC1 to prohibit HDAC1 degradation.

Conclusion

This study revealed that GSDMD was upregulated in AD lesions and that GSDMD regulated keratinocytes via epigenetic modification, which might provide potential therapeutic targets for AD.

Introduction

Atopic dermatitis (AD) is one of the most common chronic inflammatory skin diseases and is characterized by severe pruritus and relapsing eczematous lesions. It is the leading non-fatal skin disease health burden and puts patients at a higher risk of food allergies, asthma, and allergic rhinitis (Weidinger & Novak, 2016).

According to a recent study, atopic dermatitis has a specific microbiome composition that can distinguish it from psoriasis and healthy volunteers. Staphylococcus aureus, the dominant germ in AD (Fyhrquist et al., 2019), stimulates IgE production and allergic inflammatory responses (Patrick et al., 2021) and keratinocyte IL-1 expression to activate pyroptosis, a Caspase1-dependent form of inflammatory cell death (Pastar et al., 2021). According to the National Center for Biotechnology Information (NCBI) Gene Expression Omnibus (GEO) database, three expression profiles of AD patients (GDS4491, GDS2381, and GDS3806) showed gasdermin D (GSDMD) mRNA expression in AD lesions was higher than that in psoriasis lesions or healthy individuals.

A previous study showed that the expression tendency of human keratinocytes’ terminal differentiation results in a downregulation of the pyroptotic pore-forming GSDMD, which are both mediators of cornification and suppressors of pyroptosis, maintaining the homeostasis of the epidermis (Lachner et al., 2017). Filaggrin (FLG) cross-links to cornified envelopes (CEs), the composition of the epidermal barrier (Lee, 2020). Accumulating evidence of AD pathogenesis has shown that GSDMD-dependent pyroptosis switches the complex immune-mediated inflammatory response and down-regulates the expression of epidermal proteins like filaggrin (FLG), loricrin (LOR), and involucrin (IVL; Li et al., 2021a). However, the mechanisms underlying the interaction between pyroptosis-related genes and keratinocyte differentiation remain unknown.

A previous study in vascular endothelial cells suggested that Histone Deacetylase11 (HDAC11) decreased the acetylation levels of the ETS-related gene (ERG) to promote both the NLR family and the pyrin domain containing3 (NLRP3)/Caspase1/GSDMD pathways that lead to pyroptosis (Yao et al., 2022). Another study found that HDAC1 and HDAC2 depletion caused increased histone acetylation and keratinocyte proliferation in the mouse epidermis, resulting in hyperplasia and thicker epidermis (Winter et al., 2013). Another study found an increase in the expression of HDAC1 in PBMC from severe asthmatics (Pniewska-Dawidczyk et al., 2021). At the same time, asthma patients exhibited higher IL-4 mRNA levels than healthy controls (Hou et al., 2016). The results indicated that epigenetic modification like HDAC histone acetylation regulates inflammation and cornification concurrently. Thus, the possible mechanism pathway in regulating HDAC-pyroptosis-keratinocyte differentiation is still unknown. Potassium Channel Tetramerization Domain Containing 6 (KCTD6) has been shown to inhibit HDAC1 by ubiquitin-dependent degradation (De Smaele et al., 2011). In this way, the target gene is indirectly modulated through histone acetylation and, therefore, its transcriptional activity (Heride et al., 2016).

This study applied the bioinformatic method to analyze GEO data (Clough & Barrett, 2016) to explore the molecular mechanism of pyroptosis-related genes in AD. GSDMD was predicted to be upregulated in AD lesions and inhibit human keratinocyte differentiation by reducing FLG expression. It was also hypothesized that HDAC1 would decrease FLG expression by reducing histone acetylation at the FLG promoter. GSDMD blocked the interaction of KCTD6 and HDAC1 to prohibit HDAC1 degradation. Therefore, this study investigated the role of the Caspase1/GSDMD pathways in keratinocyte differentiation in AD and epigenetic mechanisms underlying histone acetylation and ubiquitination in keratinocytes.

Materials and Methods

Source of microarray data

GSE32924 contained 33 RNA expression profile samples: eight healthy control (HC) samples and 25 AD samples, which included 13 atopic dermatitis lesional skin (AL) and 12 atopic dermatitis non-lesional skin (ANL) samples.

Identification of DEGs and DEPRGs

The differential expression genes (DEGs) among the AD and control samples were identified via the GEO2R tool (Barrett et al., 2013), with |log2 FC| > 1 and p < 0.05 as the cutoff. The 161 pyroptosis-related genes (PRGs) were searched and obtained from the GeneCards database (https://www.genecards.org/) (Peng et al., 2022). Consistent genes between DEGs and PRGs were identified as differently expressed pyroptosis-related genes (DEPRGs) using the Draw Venn Diagram tool (http://bioinformatics.psb.ugent.be/webtools/Venn/).

Analysis of GO and KEGG pathway enrichment for DEPRGs

Gene Ontology (GO) enrichment analysis (which included MF (molecular function), BP (biological process), and CC (cellular component) and Kyoto Encyclopedia of Genes and Genomes (KEGG) pathway analysis were executed via the Metascape database (https://metascape.org/gp/index.html#/main/step1) (Zhou et al., 2019).

Establishment of protein-protein interaction (PPI) network

The STRING (https://string-db.org) online database tool was used to establish the PPI network (Szklarczyk et al., 2021) and the Cytoscape software was then used to visualize the PPI network and analyze the interactions of the 27 DEPRGs (Doncheva et al., 2019). The Molecular Complex Detection (MCODE) plug-in was used for the module analysis of the PPI network and the cytoHubba tool was used for identifying the hub genes.

The pyroptosis-related proteins and epidermal barrier-related proteins were assessed using STRING with 0.400 moderate confidence as the minimum interaction score.

Gene set enrichment analysis and correlation analysis between DEPRGs and epithelial differentiation genes

The gene set enrichment analysis (GSEA) tool (Subramanian et al., 2005) was used for exploring the molecular signaling pathway in which GSDMD might be involved in AD. The pathway enrichment analysis utilized the c2.cp.kegg.v7.3.symbols.gmt gene sets. A false discovery rate q-value < 0.01 was regarded as different. The heatmap of correlation between the expression of GSDMD and tight junction related core genes was created using Xiantao (xiantao.love/products/apply/).

Ethics approval and informed consent

The samples of atopic dermatitis, psoriasis, and normal skin tissues used in this study were approved by the Ethics Committee of the Guangzhou Women and Children’s Medical Center (Approval No: 2022272A01). An informed consent form, which was approved by the ethics committee, was signed by all participating patients before tissue samples were collected. This study complied with the Declaration of Helsinki. All methods and experimental protocols were carried out following all applicable guidelines and regulations.

Samples of patients and normal human skin tissue

Six atopic dermatitis, six psoriatic lesion, and six normal human skin tissue samples were obtained from the Department of Pathology, the Guangzhou Women and Children’s Medical Center. All the samples were histologically confirmed as atopic dermatitis, psoriatic, and normal human tissues by pathologists.

Histopathological analysis and immunohistochemistry

Representative psoriasis, atopic dermatitis, and healthy control skin tissues were embedded in paraffin after being treated in 4% paraformaldehyde. Toluidine blue, hematoxylin, and eosin (H&E) were used to stain the sections. Skin tissue samples were sliced into 4 µm-thick sections using a microtome, mounted on slides, and left to dry overnight at 37 °C. Slides were kept at room temperature until needed.

Slices of skin tissue were deparaffinized using gradient ethanol, followed by antigen retrieval and GSDMD immunohistochemical staining. The mouse anti-GSDMD monoclonal antibody (1:1,000; Proteintech, Rosemont, IL, USA) was applied to the slices and then left on the slices overnight at 4 °C. The immunohistochemistry kit’s instructions were followed precisely for performing the diaminobenzidine staining. Mean density was estimated using the analysis tool Image J. QuPath, a program for digital pathology, was utilized to count the positive cells in epidermis (Bankhead et al., 2017).

Cell culture and resuscitation

Normal Human Epidermal Keratinocytes (NHEK) cell lines supplied by Hechuang Bio (Beijing, China) were grown at 37 °C in an incubator with 5% CO2 in Dulbecco’s Modified Eagle’s medium (DMEM) (Lonza, Walkersville, MD, USA), 10% bovine fetal serum (Gibco, Waltham, MA, USA), and 1% penicillin-streptomycin. The cells were harvested and stored at −80 °C. Cells were resuscitated and used within six months of resuscitation.

Plasmid construction and transfection

A PCR using Homo sapiens tissue cDNA as a template amplified the full-length coding DNA sequences (CDS) of Human GSDMD (NM001166237.1) or HDAC1 (NM004964.3). The following primers (Sangon, Shanghai, China) were used: 5′-TTGGTACCGAGCTCGGATCCGCCACCATGGGGTCGGCCTTTGAGCG-3′, 5′-CATCGTCTTTGTAGTCCTCGAGTCACTAGTGGGGCTCCTGGCTCA-3′ and 5′-ATGGCGCAGACGCAGGGCAC-3′, 5′-GGCCAACTTGACCTCCTCCTT-3′.

The PrimeScript II 1st Strand cDNA Synthesis Kit was used to create cDNA from 7 ul of total RNA (TaKaRa, Shiga, Japan). According to the standard protocol of SYBR Premix Ex Taq II (TaKaRa, Shiga, Japan), PCR cycles were as follows: 95 °C for 5 min, followed by 30 cycles of 95 °C for 10 s, 56–58 °C for 15 s, and 72 °C for 2 min. The PCR fragment was then purified and subcloned into the pcDNA3.1 eukaryotic expression vector (Life). For gene transfection, the vectors were named pcDNA3.1-GSDMD and pcDNA3.1-HDAC1. Plasmids were transfected into NHEK cells using Lipofectamine 2000 (Invitrogen, Carlsbad, CA, USA), as directed by the manufacturer.

Western blot assay

The entire amount of protein obtained from the cell was separated by SDS-PAGE and transferred to a PVDF membrane, which was then treated in a TBST solution containing 5% milk to prevent the protein from binding to the antibody. The membrane was then treated with the particular antibody overnight at 4 °C, with -actin serving as the internal reference. The following day, after an hour of incubation at room temperature with the secondary antibody, the membrane was developed on an ECL detection system (BeyoECL Moon; Beyotime, Jiangsu, China) and an imaging system. Primary antibodies used for the western blot assay include: Anti-cleaved N-terminal GSDMD antibody (1:1,000, Abcam, ab215203); Anti-FLG antibody (1:1,000, Enogene, E11-0480B); Anti-K1 antibody (1:1,000, Uscn, PAA492Hu01); Anti-K10 antibody (1:10,000, Uscn, PAB691Hu01); JUP Antibody, (Enogene, E11-0138C); FLG2 antibody (BIOSS, bs-16100R); H3K56ac Antibody (Active Motif, 39282); Anti-KCTD6 antibody (Novus, H00200845-B01P); Anti-Ubiquitin antibody (Proteintech, 10201-2-AP); Anti-GAPDH antibody (1:10,000, Kangchang bio Shanghai, KC-5G5).

Immunofluorescence

NHEK cells were plated on glass and fixed in 4% paraformaldehyde solution for 30 min before being permeabilized in 0.2% Triton X-100 solution for 5 min at room temperature.

Cells were then rinsed three times in PBS, blocked for 30 min in 10% goat serum, and incubated overnight at 4 °C with human GSDMD primary antibodies (1:200, Cell Signaling, #96458), HDAC1 (1:200; Proteintech, Rosemont, IL, USA), and KCTD6 (1:200; Novus, St. Louis, MO, USA).

The cells were then washed and incubated for one hour at room temperature with a goat anti-rabbit Cy3-conjugated secondary antibody (Servicebio, Wuhan, China) or a goat anti-mouse FITC secondary antibody (Servicebio, Wuhan, China). DAPI (Servicebio, Wuhan, China) was used for nuclear counterstaining, and an automatic digital slide scanning and analysis system was used to scan and analyze the samples (Leica, Weztlar, Germany).

Quantitative reverse transcription PCR

Trizol was used to extract total RNA from NHEK cells (Invitrogen, Carlsbad, CA, USA). The PrimeScript II 1st Strand cDNA Synthesis Kit was used to create cDNA from 2 g of total RNA (TaKaRa, Shiga, Japan). The GAPDH gene was used as an internal control. According to the standard protocol, the mRNA expression was determined using qPCR and a SYBR Premix Ex Taq (Tli RNaseH Plus) (TaKaRa, Shiga, Japan). Each gene’s expression was quantified by measuring cycle threshold (Ct) values, normalized to GAPDH or GAPDH mRNA, and calculated using the 2-Ct method. Table 1 shows the primer sequences used for qPCR.

Table 1 Sequences of primers used for qPCR.

Gene		Primer sequence	Product	
GSDMD	F	AGACACAGAAGGAGGTGGA	194	
	R	GACGTCCAAGTCAGAGTCAATAA		
HDAC1	F	GGCAAGTGCTGTGAAACTTAAT	158	
	R	GCACCCTCTGGTGATACTTTAG		
FLG	F	CACTCATGAACAGTCTGAGTCC	210	
	R	CCTGAGTGTCCAGAGCTATCTA		
GAPDH	F	AACGGATTTGGTCGTATTGGG	207	
	R	CCTGGAAGATGGTGATGGGAT		

Knockdown of Caspase1 and HDAC1 by siRNA

Hechuang Bio synthesized the siRNAs that recognize HDAC1 and Caspase1 (Guangzhou, China). HDAC1’s siRNA target sequence is GCCGGUCAUGUCCAAAGUAdTdT, and Caspase1’s siRNA target sequence is GUACAGCGUAGAUGUGAAAdTdT. Real-time PCR was used to evaluate each siRNA-induced gene knockdown. NHEK cells were grown and transfected with Lipofectamine 2000 (Invitrogen, Carlsbad, CA, USA) according to the manufacturer’s protocol for siRNA transfection.

Chromatin immunoprecipitation and qPCR

A chromatin immunoprecipitation (ChIP) kit (Hechuang Biotechnology, Guangzhou, China) was used to perform a chromatin immunoprecipitation assay to determine whether HDAC1 binding to the FLG promoter region inhibits FLG expression. NHEK cells grown in a 15-cm dish were transfected with 1 ug of HDAC1 and 1 ul of Lipofectamine 2000. Cells were extracted and formaldehyde-fixed 48 h after transfection. The DNA in the cells was sonicated into 200–600 bp fragments. For both control and treatment cells, immunoprecipitation was done overnight using 0.1 ml of sheared chromatin, anti-HDAC1 antibody (2 ug, 10197-1-AP; Proteintech, Rosemont, IL, USA), and H3K56ac antibody (2 ug; Active Motif, Carlsbad, CA, USA). Real-time PCR was used to determine the quantities of the specified DNA fragments after the DNA was separated from immunoprecipitated chromatin.

Primers were created to cover the FLG promoter region, explicitly targeting HDAC1 and acetylated H3. The primer sequences are listed in Table 2. Each reaction employed 2 ul of HDAC1-enriched cDNA from untreated and treated cells (each sample had triplicates). TB Green Premix Ex Taq was used to amplify the cDNA (TaKaRa, Shiga, Japan). The ChIP-qPCR data were represented as the relative enrichment fold change of the HDAC1-OE group relative to the Ctrl group, with three replicates for each group.

Table 2 Sequences of primers used for qPCR after ChIP.

Gene		Primer sequence	bp	
FLG promoter 1	F	CCTCCTTCTCTTTGTCTTTCTGT	120	
	R	CCAGTCTCTCTTTCTCTCTCTCT		
FLG promoter 2	F	GTCTCATGTTCATCTCCCTGTAA	130	
	R	AGAAGGACCAATGTGGTATGG		
FLG promoter 3	F	CACTCTCTCAATGCTACCTTCTT	108	
	R	CAAAGTACCTATGTCTGGGTGAA		
GAPDH	F	AAAAGCGGGGAGAAAGTAGG	212	
	R	AAGAAGATGCGGCTGACTGT		

Dual-luciferase reporter assay

The FLG promoter was initially cloned into the PGL4.10 basic luciferase reporter vector (Hechuang, Guangzhou, China). NHEK cells were seeded into 96-well plates and grown for 24 h. Then, plasmids containing FLG promoter luciferase reporter vector were co-transfected following the same methods used for the experimental groups. After co-transfection, the FLG promoter plasmid, 100 ng of the PGL4.10-FLG expression vector, and 1 µg of the Renilla reporter plasmid (Hechuang, Guangzhou, China) were normalized in each well. Using a Dual-Luciferase Reporter Gene Analysis System (Promega, Madison, WI, USA), the luciferase activity of PGL4.10-FLG-Pro was measured after 24 h of incubation.

Co-immunoprecipitation assays

The co-immunoprecipitation (Co-IP) Kit (Abison, Guangzhou, China) protocol was followed for the co-immunoprecipitation assays, and the cells required for the experiment were cultivated and gathered. The cells were given a single wash with 1× PBS after being freed from the DMEM. The cell plate (10 cm) was then filled with 0.5 ml of ice-cold lysis buffer, which was then incubated for at least 5 min on ice. The cells were then scraped off and put in a fresh EP tube. The subsequent experiment was conducted according to the manufacturer’s instructions. Approximately 160 μg GSDMD or HDAC1 protein sample was incubated with 2 μg GSDMD or HDAC1antibody. The interaction of the HDAC1 and KCTD6 or GSDMD and KCTD6 proteins were then detected and analyzed using an immunoprecipitation (IP) immunoblotting (IB) test. Approximately 160 μg GSDMD overexpression or NHEK protein sample was incubated with 2 μg HDAC1 antibody. The interaction of the HDAC1 and KCTD6 proteins were then detected and analyzed as above. KCTD6 overexpression or NHEK protein sample was incubated with 2 μg Ubiquitin antibody to detect the ubiquitination and subsequent degradation of HDAC1. The HDAC1 proteins were then detected and analyzed using IB.

Statistical analysis

All data for the statistical analysis were obtained from three independent replications and presented as mean ± SD. Statistical significance was determined using the Student’s two-tailed t test, Mann-Whitney test, or one-way ANOVA with the Bonferroni post-test for single or multiple comparisons, as appropriate, using SPSS 17.0 and GraphPad Prism 5. A p-value of 0.05 was deemed statistically significant.

Results

Analysis of DEPRGs and hub genes in AD

To explore the molecular mechanism of pyroptosis-related genes in AD, 2,198 DEGs were identified in the GSE32924 dataset, as shown in Fig. 1A. Then, 27 congruous DEPRGs were determined with an integrated bioinformatics analysis, including 11 upregulated genes (IRF3, GSDMC, SERPINB1, BIRC3, CAPN1, IRF1, IL32, GSDMD, MKI67, CD274, GZMB) and 16 downregulated genes (MALAT1, CTSV, TP63, ORMDL3, TP53, TXNIP, PRDM1, STK4, IL1RN, ATF6, IL18, PANX1, CASP1, PECAM1, DHX9, IFI16; Fig. 1B).

Figure 1 Analysis of the DEPRGs and GSEA of GSDMD in AD.

(A) The 21 AL DEPRGs and 17 ANL DEPRGs of AD in GSE32924. (B) The PPI network of 27 DEPRGs. (C) The10 hub genes of DEPRGs in AD. (D) Enriched ontology clusters created by metascape.org. (E) The GSEA of GSDMD. (F) The heatmap of correlation between the GSDMD and tight junction. The consistent genes between DEGs and PRGs were identified as DEPRGs.

To identify hub genes involved in DEPRGs, the STRING database, which includes 27 nodes and 62 edges, was used to establish the PPI network of the 27 DEPRGs (Fig. 1B). The hub genes were then further analyzed by cytoHubba, and ten genes were identified as hub genes in AD: IRF3, IRF1, GSDMD, CD274, GZMB, TP53, PRDM1, IL18, CASP1, and IFI16 (Fig. 1C).

Functional analysis of DEPRGs and GSEA of GSDMD in AD

To determine the functions of the DEPRGs identified, a KEGG pathway enrichment was performed. Results revealed that the DEPRGs were associated with the NOD-like receptor signaling pathway, pyroptosis, and the epithelial cell differentiation signaling pathway (Fig. 1D).

Of the hub genes identified for AD, both GSDMD and TP53 were found to also be DEPRGs in AL and ANL, and were thus chosen for further analysis. GSEA results showed that GSDMD contributed to adherens junction, arachidonic acid metabolism, gap junction, retinol metabolism, and the epithelial tight junction signaling pathway (Fig. 1E). GSDMD, but not TP53, was found to be statistically and negatively associated with epithelial tight junction core gene expression (Fig. 1F). Therefore, GSDMD was identified as a possible essential hub gene for AD, and subsequent experiments further investigated the role of GSDMD.

GSDMD is upregulated in AD lesions

A PPI network analysis was performed using STRING to assess the interaction between pyroptosis-related proteins and epidermal barrier-related proteins. Results indicated that GSDMD directly interacted with JUP and FLG2, and indirectly interacted with FLG, LOR, and IVL (Fig. 2A).

Figure 2 GSDMD is upregulated in AD lesions.

(A) Protein-protein interaction (PPI) network analysis shows GSDMD connects with JUP, FLG2, FLG, LOR, and IVL. (B and C) Immunohistochemistry showed expressions of GSDMD (staining dark brown) in AD patient lesions, psoriasis lesions, and normal human skin. The yellow line area represents epidermis area analyzed and the positive cells is marked in red circles by QuPath. (B) HC, healthy control; AD, atopic dermatitis; Ps, psoriasis. (C) The bar graph of the three groups represents quantitative GSDMD expression by optical density (OD) value for immunohistochemistry.

To determine differences in GSDMD protein expression between various skin diseases, GSDMD was detected by immunohistochemistry in HC skin, AD lesions, and psoriasis lesions. We used the Image J to access staining density and the digital pathology software QuPath to count the number of positive cells in epidermis. Results showed that the expression of the GSDMD protein (dark brown staining) was upregulated in AD lesions and psoriasis lesions (Figs. 2B and 2C), and there were more positive cells (in red circles) in the psoriasis epidermis (38%) than in AD epidermis (19.68%) and HC epidermis (3.6%) (Fig. 2B).

GSDMD suppresses human keratinocyte differentiation

To determine the role of GSDMD in human keratinocytes, GSDMD expression vector was constructed to overexpress GSDMD in NHEK cells. Transfection of the GSDMD expression vector significantly enhanced the expression of the GSDMD protein in NHEK cells (Fig. 3A). GSDMD overexpression also reduced the expression level of terminal differentiation proteins, including JUP, FLG, and FLG2 in NHEK cells (Fig. 3B). These results indicate that GSDMD inhibits human keratinocyte differentiation.

Figure 3 GSDMD suppresses human keratinocyte differentiation.

(A) The level of GSDMD protein in NHEK cells transfected with blank vector or GSDMD expression vector. (B) The protein levels of JUP, FLG, and FLG2 in NHEK cells transfected with blank vector or GSDMD expression vector. NC, negative control. **p < 0.01, ***p < 0.001.

Caspase1-activated GSDMD inhibits human keratinocyte differentiation by reducing FLG expression

Caspase1 plays a crucial role in GSDMD activation (Shi et al., 2015), so the effect of Caspase1 on GSDMD in human keratinocytes was explored. SiRNA (Caspase1 siRNA 2) was used to silence Caspase1 (Fig. 4A). Results of WB showed that silencing Caspase1 led to the reduction of activated GSDMD (N-GSDMD) in NHEK cells (Fig. 4B), suggesting that Caspase1 activates GSDMD in human keratinocytes.

Figure 4 Caspase1-activated GSDMD inhibits human keratinocyte differentiation by reducing FLG expression.

(A) The level of Caspase1 mRNA in NHEK cells transfected with NC siRNA or Caspase1 siRNAs. (B) The protein levels of N-GSDMD, FLG, K1, and K10 in NHEK cells transfected with NC siRNA or Caspase1 siRNA. NC, negative control. **p < 0.01, ***p < 0.001, ****p < 0.0001.

Markers of human keratinocyte differentiation were analyzed by WB to confirm the effect of Caspase1 on cornification. In contrast to activated GSDMD, silencing Caspase1 increased the expression of FLG, Keratin 1 (K1), and Keratin 10 (K10) in NHEK cells (Fig. 4B). GSDMD overexpression reduced FLG expression (Fig. 3A), indicating that GSDMD activated by Caspase1 suppresses human keratinocyte differentiation by reducing FLG expression.

GSDMD reduces FLG expression through HDAC1 in human keratinocytes

Because HDAC1 promotes Caspase1-GSDMD pyroptosis and decreases keratinocyte proliferation, the effect of HDAC1 regulation was also assessed (Winter et al., 2013; Yao et al., 2022). Consistent with GSDMD overexpression, HDAC1 overexpression by transfection of the HDAC1 expression vector also decreased the mRNA (Fig. 5A) and protein levels (Fig. 5B) of FLG in NHEK cells. Conversely, silencing HDAC1 increased the mRNA (Fig. 5A) and protein levels (Fig. 5B) of FLG in NHEK cells. Silencing HDAC1 also reversed the suppression effect of GSDMD overexpression on FLG expression (Figs. 5A and 5B). These results suggest that GSDMD decreases FLG expression through HDAC1 in human keratinocytes.

Figure 5 GSDMD reduces FLG expression via HDAC1 in human keratinocytes.

(A) The mRNA levels of GSDMD, HDAC1, and FLG in NHEK cells. (B) The protein levels of GSDMD, HDAC1, and FLG in NHEK cells. (C) Immunoprecipitated chromatin was analyzed by qPCR for the FLG gene promoter in NHEK cells after ChIP using HDAC1 antibody. (D) Immunoprecipitated chromatin was analyzed by qPCR for the FLG gene promoter in NHEK cells after the ChIP using H3K56ac antibody. NC, negative control. **p < 0.01, ***p < 0.001.

As HDAC1 reduces histone acetylation at the gene promoter to inhibit gene expression, the interaction of HDAC1 and FLG promoter was detected by ChIP using HDAC1 antibody in NHEK cells. ChIP results showed that HDAC1 was indeed associated with the FLG promoter in NHEK cells (Fig. 5C). Moreover, HDAC1 overexpression enhanced the interaction of HDAC1 and the FLG promoter in NHEK cells (Fig.5C).

The histone acetylation at the FLG promoter was also detected by ChIP using an H3K56ac antibody. Results revealed that HDAC1 overexpression significantly decreased histone acetylation at the FLG promoter in NHEK cells (Fig.5D). These results suggest that HDAC1 decreases FLG expression by reducing histone acetylation at the FLG promoter in human keratinocytes.

GSDMD prohibits HDAC1 degradation by blocking the interaction of KCTD6 and HDAC1 in human keratinocytes

Results of qPCR, WB, and IHC indicated that GSDMD overexpression also increased mRNA and protein levels of HDAC1 in NHEK cells (Figs. 5A, 5B and 6A), suggesting GSDMD enhances HDAC1 expression in human keratinocytes.

Figure 6 GSDMD prohibits HDAC1 degradation by blocking the interaction of KCTD6 and HDAC1 in human keratinocytes.

(A) Images of IF performed by HDAC1 antibody in NHEK cells transfected with blank vector or GSDMD expression vector. (B) Representative images of WB for KCTD6 and silver staining following Co-IP performed by GSDMD or HDAC1 antibody in NHEK cells. (C) Images of IF performed by KCTD6 antibody combined with GSDMD or HDAC1 antibody in NHEK cells. (D) Representative images of WB for KCTD6 and silver staining following Co-IP performed by HDAC1 antibody in NHEK cells transfected with blank vector or GSDMD expression vector. (E) Representative images of WB for HDAC1 and silver staining following Co-IP performed by GSDMD antibody in NHEK cells transfected with blank vector or KCTD6 expression vector. (F) The protein level of HDAC1 in NHEK cells transfected with blank vector, KCTD6 expression vector, or GSDMD expression vector combined with KCTD6 expression vector. NC, negative control; vec, expression vector. ***p < 0.001, ****p < 0.0001.

The mechanism behind GSDMD’s ability to regulate HDAC1 was then explored. Previous studies demonstrated that the GSDMD protein associates with the KCTD6 protein in human cells (Rolland et al., 2014) and that KCTD6 can induce HDAC1 degradation (De Smaele et al., 2011). Co-IP results revealed that the GSDMD protein was also associated with the KCTD6 protein and that the KCTD6 protein interacted with the HDAC1 protein in NHEK cells (Fig. 6B). IHC further confirmed colocalization of the GSDMD protein and the KCTD6 protein (Fig. 6C), or the KCTD6 protein and the HDAC1 protein (Fig. 6D), in NHEK cells. GSDMD overexpression also blocked the interaction of the KCTD6 protein and the HDAC1 protein in NHEK cells (Fig. 6D). Conversely, KCTD6 overexpression by transfection of KCTD6 expression vector reduced the interaction of the GSDMD protein and the HDAC1 protein in NHEK cells (Fig. 6E). These findings suggest that the GSDMD protein competes with the KCTD6 protein to bind to the HDAC1 protein in human keratinocytes.

The effect of KCTD6 on HDAC1 was also identified. Results of WB showed that GSDMD overexpression increased HDAC1 protein levels in NHEK cells. Conversely, KCTD6 overexpression neutralized the effect of GSDMD overexpression on HDAC1 protein level (Fig. 6F), suggesting that GSDMD blocks the interaction of KCTD6 and HDAC1 to prohibit HDAC1 degradation in human keratinocytes.

Discussion

This study revealed that GSDMD might be the most crucial hub gene for AD. GSDMD was upregulated in AD lesions and inhibited human keratinocyte differentiation by reducing FLG expression. Mechanistically, GSDMD activated by Caspase1 decreased FLG expression via HDAC1. HDAC1 decreased FLG expression by reducing histone acetylation at the FLG promoter. GSDMD blocked the interaction of KCTD6 and HDAC1 to prohibit HDAC1 degradation (Fig. 7).

Figure 7 Schematic diagram of molecular mechanisms for this study.

The present study revealed that GSDMD was upregulated in AD lesions and inhibited human keratinocyte differentiation by reducing FLG expression. Mechanistically, GSDMD activated by Caspase1 reduced FLG expression via HDAC1. HDAC1 decreased FLG expression by reducing histone acetylation at the FLG promoter. In addition, GSDMD blocked the interaction of KCTD6 and HDAC1 to prohibit HDAC1 degradation.

GSDMD is known to be closely associated with pyroptosis (Burdette et al., 2021; Shi et al., 2015; Zuo et al., 2021), but the role of GSDMD in cell differentiation remains unclear. Only two recent studies have indicated that GSDMD-mediated pyroptosis inhibits osteoblast differentiation (Li et al., 2022; Zhang & Wei, 2021). The latest research has shown that after induction of AD-like skin lesions, mice lacking GSDMD exhibited relieved AD signs and symptoms in terms of reduced skin thickness, scarring and scratching behavior compared to wild-type mice. This was associated with decreased infiltration of inflammatory cells, reduced epidermal thickness, and decreased expression levels of IgE, IL-4, IL-1β and IL-18 (Lu et al., 2023). As a possible role of the GSDMD-dependent pyroptosis in keratinocyte under AD-like inflammation has not been fully addressed, we processed GSDMD overexpression in keratinocytes. The current study is the first to identify the role of GSDMD in human keratinocyte differentiation in AD. In this study, the terminal epithelial differentiation proteins, FLG, FLG2, JUP, K1, and K10, were attenuated.

Numerous studies have indicated that activation of the NLR family, pyrin domain containing 3 (NLRP3) inflammasome, including Caspase1, is essential for GSDMD activation by the cleavage of GSDMD (Gao et al., 2021; Li et al., 2021b; Shi et al., 2015). The NLRP3 inflammasome is activated in AD (Kim et al., 2022; Liu et al., 2022) and induces GSDMD activation in AD (Li et al., 2021a). Therefore, the GSDMD of human keratinocytes is likely triggered by the NLRP3 inflammasome in AD. The current study showed Caspase1 activates GSDMD and suppresses FLG expression.

As a histone deacetylase, HDAC1 usually reduces histone acetylation at the gene promoter to inhibit target gene expression (Dai et al., 2022; Moreno-Yruela et al., 2022). However, the effect of HDAC1 on FLG expression is largely still being determined. Only one study has demonstrated that HDAC1 decreases FLG expression in keratinocytes under tumor necrosis factor-alpha (TNFα) and interferon-gamma (IFNγ) stimulation (Ahn et al., 2022). The present study verified that the overexpression of GSDMD led to the attenuation of FLG. When GSDMD was overexpressed and HDAC1 was simultaneously silenced, the expression of FLG was increased, indicating that GSDMD inhibited FLG expression by promoting HDAC1. In HDAC1 overexpression keratinocytes, a ChIP test using the HDAC1 antibody verified that FLG expression was suppressed through HDAC1 binding to the FLG promoter. A separate ChIP test using the H3k56ac antibody found that the level of acetylation histone H3 binding to the promoter of FLG was decreased, leading to the down-regulation of FLG transcription. The results of this study indicate that HDAC1 histone acetylation regulates the keratinocyte differentiation of AD through FLG expression.

Previous studies have revealed that the KCTD6 protein binds to the HDAC1 protein and subsequently induces ubiquitination-dependent degradation of HDAC1 proteins (De Smaele et al., 2011; Pirone et al., 2013). Thus, KCTD6 should also cause the degradation of the HDAC1 protein by ubiquitination. The CO-IP assay conducted in this study confirmed the interaction between KCTD6 and both GSDMD and HDAC1. The upregulation of KCTD6 promoted the ubiquitination and subsequent degradation of HDAC1, leading to a significant reduction in its detectable levels. It is plausible that GSDMD competes with HDAC1 for binding to KCTD6. The rescue experiment provides evidence that GSDMD upregulates HDAC1 levels through the inhibition of KCTD6. Co-overexpression of both GSDMD and KCTD6 led to an excessive binding of KCTD6 to GSDMD, resulting in an abundance of KCTD6 (including endogenous KCTD6) that could bind to and degrade HDAC1. Consequently, the overexpression of KCTD6 diminished the protein level of HDAC1 and reinstated FLG expression, suggesting that KCTD6 may have therapeutic potential for treating FLG depletion and the activation of pyroptosis in AD.

The results of this study indicate that GSDMD blocked the interaction of KCTD6 and HDAC1 to prohibit KCTD6-induced degradation of HDAC1 in human keratinocytes. The presents study is the first to reveal the role of GSDMD on protein degradation.

Future studies should determine GSDMD-induced pyroptosis of human keratinocytes, and the findings of the current study should be confirmed through in vivo experiments.

Conclusions

The results of this study indicate that GSDMD might be an essential hub gene for AD. GSDMD was upregulated in AD lesions and inhibited human keratinocyte differentiation by reducing FLG expression. Mechanistically, GSDMD activated by Caspase1 reduced FLG expression via HDAC1. HDAC1 decreased FLG expression by reducing histone acetylation at the FLG promoter. In addition, GSDMD blocked the interaction of KCTD6 and HDAC1 to prohibit HDAC1 degradation. This study revealed the mechanism of GSDMD in regulating AD, which might provide potential therapeutic targets for AD.

Supplemental Information

Supplemental Information 1 Raw data for Figures 4 and 5.

Click here for additional data file.

Supplemental Information 2 Raw images of Figure 3A.

Click here for additional data file.

Supplemental Information 3 Raw images of Figure 3B.

Click here for additional data file.

Supplemental Information 4 Raw data of Figure 4B: FLG and GSDMD.

Click here for additional data file.

Supplemental Information 5 Raw data of Figure 4B: K1 and K10 and GAPDH.

Click here for additional data file.

Supplemental Information 6 Raw data of Figure 5B: GAPDH and FLG.

Click here for additional data file.

Supplemental Information 7 Raw data of Figure 5B: HDAC1 and GSDMD.

Click here for additional data file.

Supplemental Information 8 Raw data of Figure 6BDE.

Click here for additional data file.

Supplemental Information 9 Raw data of Figure 6F.

Click here for additional data file.

Supplemental Information 10 IHC AD *20.

Click here for additional data file.

Supplemental Information 11 IHC HC*20.

Click here for additional data file.

Supplemental Information 12 IHC Ps*20.

Click here for additional data file.

Supplemental Information 13 IHC HC*10.

Click here for additional data file.

Supplemental Information 14 IHC HC*4.

Click here for additional data file.

Supplemental Information 15 IHC AD*4.

Click here for additional data file.

Supplemental Information 16 IHC Ps*4.

Click here for additional data file.

Supplemental Information 17 IHC AD*10.

Click here for additional data file.

Supplemental Information 18 IHC Ps*10.

Click here for additional data file.

Additional Information and Declarations

Competing Interests

Author Contributions

Human Ethics

Data Availability

The authors declare that they have no competing interests.

Yi Zhong conceived and designed the experiments, performed the experiments, analyzed the data, prepared figures and/or tables, authored or reviewed drafts of the article, and approved the final draft.

Taoyuan Huang performed the experiments, authored or reviewed drafts of the article, and approved the final draft.

Xiaoli Li performed the experiments, prepared figures and/or tables, authored or reviewed drafts of the article, and approved the final draft.

Peiyi Luo analyzed the data, authored or reviewed drafts of the article, and approved the final draft.

Bingjun Zhang conceived and designed the experiments, analyzed the data, prepared figures and/or tables, authored or reviewed drafts of the article, and approved the final draft.

The following information was supplied relating to ethical approvals (i.e., approving body and any reference numbers):

The Ethics Committee of the Guangzhou Women and Children’s Medical Center approved the study (Approval No: 2022272A01).

The following information was supplied regarding data availability:

The raw data are available in the Supplemental Files.

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
