# Peer review of "GSDMD suppresses keratinocyte differentiation by inhibiting FLG expression and attenuating KCTD6-mediated HDAC1 degradation in atopic dermatitis"

_PeerJ, doi:10.7717/peerj.16768_

## Round 0.1 · original submission · Major Revisions

Please address the concerns of both reviewers and amend the manuscript accordingly.

**Language Note:** The review process has identified that the English language must be improved. PeerJ can provide language editing services - please contact us at copyediting@peerj.com for pricing (be sure to provide your manuscript number and title). Alternatively, you should make your own arrangements to improve the language quality and provide details in your response letter. – PeerJ Staff

Reviewer 1 ·

Basic reporting

The figures are rather blurry, especially when being enlarged
All the abbreviations used in the study should be spelled out the first time they were used.
In addition, the formatting of the references in the text should be improved. In the current format it is very difficult to read.
In the results part, the explanation why an experiment is performed and why certain molecules are now investigated is often missing
Figure legends: please include how many replicates were done for quantification of the blots, IPs ect.
Statistical tests: if more than 2 samples are compared, one or two way ANOVA should be used instead of students T-test
The English language has to be revised extensively

Experimental design

the manuscript is interesting and scope of the journal
the research question is well defined and will fill a knowledge gap in the examination in AD.
Nevertheless the discussion of the results should be better discussed and be discussed in a broader context

Validity of the findings

See point 2: not all of the conclusions are well stated, see also comments to point 4
also not all of the data support the conclusions the authors have drawn

Additional comments

In their manuscript, Zhong et al. describe that gasdermin D is enhanced in human AD samples. In a bioinformatics approach gasdermin D was found to be negatively correlated to tight junction proteins. They then analyzed the role of gasdermin D in human keratinocytes and could show that gasdermin D activated by caspase 1 blocked Potassium Channel Tetramerization Domain Containing 6 (KCTD6), which promotes the ubiquitination and degradation of HDAC1. Blockade of KCTD6 action enabled HDAC1 to act on the filaggrin promotor, thus inhibiting filaggrin expression. In summary, gasdermin activation in AD contributes to barrier damage via filaggrin downregulation. The study is very interesting thus linking activation of keratinocytes via inflammasome/gasdermin D to barrier dysregulation via regulation of filaggrin. Nevertheless, some important questions and issues have to be addressed by the authors.

Specific points:
The authors showed that gasdermin overexpression, as well as siRNA mediated caspase-1 knockdown influences filaggrin and keratin 1/10 levels in NHEK cells. In addition, overexpression of gasderim D ehanceds HDAC1 levels and HDAC1 overexpression reduced filaggrin levels. So the caspase-1-gasderminD-HDAC1 axis is clearly shown. The contribution of KCTD6 however is unclear. KCTD6 co-immunoprecipitates with gasdermin D and HDAC1, but the way from gasdermin D/KCTD6 interaction to enhanced HDAC1 levels remains unclear. According to the scheme in Figure 7 the interaction of gasdermin D/KCTD6 should lead to more HDAC1, in Figure 6 F there is reduced HDAC1. Reduced HADAC1 levels should lead to enhanced filaggrin levels. How do the authors explain this?
The materials and methods section is rather short and not detailed written. It is not clear which antibodies were used for which experiment and which cells are when used.
In lane 252 the “relevant plasmids” should be mentioned with names
Minor point: the term “µm” is missing in lane 164 and 254

Figure 2: skin histology. The specific staining for gasdermin D is not really visible. Maybe the authors should add arrows to show where gasdermin D positive cells are located. In addition, staining is seen in epidermis (AD samples) and dermis (both AD and psoriasis). How are the samples quantified? Since dermal and epidermal changes are different in AD and psoriasis, how was this done?

In Figure 5A and B two different diagrams are shown. What is the difference between the two diagrams?
Text lane 333 and 334: the silencing of….
Figure 6A and 6C: the staining are very difficult to detect since the Figures are very dark and the signals very faint.

The discussion section is also rather short and gives no information about the meaning of the results obtained in terms of AD induction and pathology or therapy options. Also, there it is not really clear what enhanced HDAC1 levels besides directly interacting with filaggrin do? Is this also important for AD pathology? What are consequences of gasdermin D activation in AD? The pathway described here is certainly only one path that is possible but all the molecules involved also feed into other pathways. This has to be discussed in more detail.

General question: gasdermin D is only upregulated in AD not in psoriasis? What is the explanation for that? Psoriasis is an inflammatory disease and the inflammasome should be activated here. What is the special association of gasdermin D with AD pathology?

Formatting of the manuscript:
The figures are rather blurry, especially when being enlarged
All the abbreviations used in the study should be spelled out the first time they were used.
In addition, the formatting of the references in the text should be improved. In the current format it is very difficult to read.
In the results part, the explanation why an experiment is performed and why certain molecules are now investigated is often missing
Figure legends: please include how many replicates were done for quantification of the blots, IPs ect.
Statistical tests: if more than 2 samples are compared, one or two way ANOVA should be used instead of students T-test
The English language has to be revised extensively

Reviewer 2 ·

Basic reporting

The article is well written and sufficient introduction is provided to set the stage for the establishing the hypothesis. The hypothesis developed using bioinformatics tools is well supported by empirical evidence to arrive at the conclusion that GSDMD may be an essential Hub gene for atopic dermatitis. This manuscript also demonstrates a plausible mechanism that may be involved in regulation of AD by GSDMD gene. This otherwise being a well written manuscript, has a few typos that have been mentioned in the additional comments.

Experimental design

The research question is well established i.e. To investigate interaction mechanism between pyroptosis-related genes and human keratinocyte differentiation.
Experimental section is well written with sufficient information needed to replicate the work.
Limitations of the study are also well identified in the manuscript.

Validity of the findings

This seems to be the first study associating GSDMD to human keratinocyte differentiation in AD. This study also showed the role of GSDMD on protein degradation for the first time. this work will potentially lead to more detailed study on GSDMD gene and will lead to better understanding its role in AD and other diseases.

Additional comments

I am happy to accept this manuscript with a few minor edits mentioned below.
Please correct GDSMD to GSDMD at the following locations.
1- Line 382
2- Figure 3
3- Figure 6
Please correct Line 188 – 5658 °C for 15 seconds.
Please add more information about the PCR reagents and the polymerase used.

---

## Round 0.2 · accepted · Accept

All concerns of the reviewers were addressed and the revised manuscript is acceptable now.

Reviewer 2 ·

Basic reporting

The language has improved significantly in the revised version. Sufficient literature is cited to put the study in context and provide enough background. Although, the figure resolution has been improved, a bit higher resolution may further enhance the quality.

Experimental design

The materials and methods section has sufficiently improved by providing the information that was missing in the previous submission. The research question is well defined and the experimental results support the hypothesis.

Validity of the findings

The current work does seem to indicate that GSDMD is an essential hub gene playing a role in regulation AD via HDAC1. Data from qPCR, WB,IHC, and CO-IP experiments, together suggests the role of GSDMD in modulating KCTD6-induced degradation of HDAC1 in human keratinocytes. There is evidence in the work presented here that overexpression of KCTD6 reduces the level of HDAC1 protein leading to increased FLG levels that regulate AD. This study suggests that KCTD6 may have therapeutic potential for treating FLG depletion and the activation of pyroptosis in AD.

Additional comments

I am happy to accept this manuscript in its current form.